# Dynamic Changes and Correlation Analysis of Polysaccharide Content and Color Parameters in Glycyrrhiza Stems and Leaves during Fermentation

**Juan Du** [1,2,3], **Yifeng Song** [1,2,3], **Xia Li** [1,4], **Na Liu** [1,2,3], **Xiaoping An** [1,2,3] **and Jingwei Qi** [1,2,3,*]

[1] College of Animal Science, Inner Mongolia Agricultural University, Hohhot 010018, China; dujuan_ddj@163.com (J.D.); songyifeng777@163.com (Y.S.); lixia2270@126.com (X.L.); liuna_dky@163.com (N.L.); xiaoping_an@163.com (X.A.)

[2] Inner Mongolia Herbivorous Livestock Feed Engineering Technology Research Center, Hohhot 010018, China

[3] Key Laboratory of Smart Animal Husbandry, Universities of Inner Mongolia Autonomous Region, Hohhot 010018, China

[4] Inner Mongolia Shengmu Animal Husbandry Co., Ltd., Hohhot 010020, China

[*] Correspondence: qijingwei@imau.edu.cn

**Abstract:** Fermentation can increase the concentration of active ingredients and improve the effectiveness of Chinese herbal medicine. The purpose of this study was to investigate the effect of solid-state fermentation (SSF) on the polysaccharide content and color of Glycyrrhiza stems and leaves, as well as to explore the potential of computer vision-based analytical chemistry for the rapid, non-destructive, and accurate quality identification of fermented herbs. The effects of different inoculation rates on the polysaccharide content and color of fermented Glycyrrhiza stems and leaves were evaluated. Subsequently, dynamic changes in the viable counts of the probiotic strains, pH values, polysaccharide content, and color of Glycyrrhiza stems and leaves were explored during the entire fermentation process. The correlations of color variables that were extracted from the images with key quality indicators of the Glycyrrhiza stem and leaf samples were verified. The results showed that with an increase in inoculation amount, the polysaccharide content demonstrated a trend of first increasing and then decreasing, which was consistent with the color parameter behavior, and the optimal inoculation amount was 0.2%. During fermentation, R, G, B, S, V, L, a*, and B* were significantly correlated with the polysaccharide content ($p < 0.01$), while the correlation of H was weak. Principal component analysis (PCA) based on color variables can effectively distinguish between different stages of fermentation. This study provides a reference for the rapid and nondestructive analysis of fermented Glycyrrhiza stems and leaves, offering a new approach to process monitoring and quality control of fermented herbs.

**Keywords:** solid fermentation; computer vision system; color space; active substance; Chinese herbal medicine

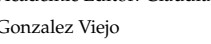


## 1. Introduction

In recent years, there has been a growing interest in natural products derived from medicinal plants as an important source of bioactive compounds for the treatment of human and animal diseases [1]. Glycyrrhiza is a perennial herb that is distributed in the northeast and eastern parts of China, including Heilongjiang, Jilin, Liaoning, Hebei, Inner Mongolia, and Shaanxi Provinces [2–4]. In traditional Chinese medicine, Glycyrrhiza is bestowed with the laudatory title of "king of medicinal materials" due to its diverse range of active substances. Polysaccharides are some of the main active ingredients of Glycyrrhiza, which have many functions such as immunity regulation, antivirus, antioxidant, and antitumor effects [5]. However, only the subterranean parts of Glycyrrhiza are utilized for medicinal purposes. The aerial portions (including leaves and stems), which account for at least

one-third of the plant's total biomass, contain a wealth of active compounds [6]. Since most of the bioactive compounds, such as polysaccharides, in Glycyrrhiza stems and leaves are trapped within complex structures containing lignin, cellulose, and proteins in different forms, their utilization is low, and they are often considered agricultural waste. Therefore, processing technologies and quality evaluation methods are needed for releasing bioactive compounds from Glycyrrhiza stems and leaves. It will be beneficial for the development of food or feed additives and for the promotion of the potential innovative applications of Glycyrrhiza by-products.

Solid-state fermentation (SSF) is an effective method to increase the content and extractability of active substances in medicinal plants [7]. Lactic acid bacteria, Bacillus, and yeast are widely used microorganisms in fermentation and possess properties that enhance important nutritional and/or sensory characteristics when inoculated in suitable quantities [8]. To be specific, microorganisms in the fermentation environment destroy the feed structure by destroying the cell wall and degrading the protein, thus releasing its active components [9]. For example, SSF can improve the bioavailability of flavonoids from dandelion, which can be utilized for the preparation of natural antioxidants, functional foods, or food additives [10]. The mulberry leaves' total flavonoid content released with SSF (72.55 mg rutin equivalents/g dry weight) was significantly higher than that of unfermented leaves (24.42 mg rutin equivalents/g dry weight) [11]. At present, food and animal feed production using probiotic fermentation has become a hot topic.

In addition, the carbonyl reaction between the free amino group of the protein and the reducing end of the carbonyl group of the polysaccharide initiates the Maillard reaction during the fermentation process, which causes the production of a blend of intermediate and/or advanced MRPs that have a yellow or brown color [12], ultimately leading to a change in color in the fermented herbal medicine product. Therefore, sensory evaluation by workers is commonly used to assess the quality of fermentation [7]. Specifically, the worker evaluates the degree of fermentation by discerning changes in color and odor within the fermenting materials. However, this method is subjective and lacks complete accuracy and reproducibility, thus making it impossible to guarantee quality standards for each batch of fermentation material. Laboratory-based chemical analysis can accurately determine the concentration of key components in a fermented sample; however, this analytical process is time-consuming, expensive, and intricate.

To obtain information on the chemical composition of food efficiently and affordably, non-invasive techniques are gaining increasing attention. Computer vision-based analytical chemistry (CV-AC) has recently emerged as a valuable tool for establishing a correlation between color properties in digital images of food and quantitative data obtained through chemical analysis methods [13,14]. Image analysis of color enables the characterization of color parameters in digital images, which are transformed into numerical values providing analytical information related to the concentration of the analyte of interest [15].

However, there have been no reports so far on the use of fermentation technology to improve the utilization rate of Glycyrrhiza stems and leaves, and no attempts have been made to analyze the correlation between the color of fermentation materials and active substances using CV-AC. Thus, this study aims to investigate the following: (1) the effect of different bacterial additions on the content of polysaccharide and color attributes; (2) dynamic changes in polysaccharide content and color; and (3) the correlation relationship between the polysaccharide content and color in Glycyrrhiza stems and leaves during fermentation. It is believed that this study will provide guidance for development of food or feed additives from Glycyrrhiza stems and leaves, promoting the potential innovative application of Glycyrrhiza by-products and laying the foundation for the standardization, informatization, and intelligent quality evaluation of the fermentation process. Correspondingly, we have added and modified the description of "the aim of fermentation" in the abstract and introduction.

## 2. Materials and Methods

### 2.1. Preparation of Fermented Samples

Fresh Glycyrrhiza stem and leaves were collected in Hohhot, Inner Mongolia, China, in July after three years of cultivation of Glycyrrhiza. The aerial parts of Glycyrrhiza were air-dried, crushed to 3–5 cm, and used as the main substrate for SSF. Glycyrrhiza stems and leaves (60%, $w/w$) were mixed with other ingredients, including maize meal (20%, $w/w$), soybean meal (10%, $w/w$), wheat bran (10%, $w/w$), and cellulase (0.5%, $w/w$), and the mixture was divided into six equal piles, i.e., inoculated with 0.1%, 0.2%, 0.4%, 0.8%, 1.6%, and 3.2% of compound probiotics (*Bacillus subtilis*, *Lactobacillus plantarum*, and *Saccharomycetes* cerevisiae mixed in a ratio of 1:1:1), respectively, for screening of the inoculum amount. While stirring evenly, sterile water was added to achieve a total moisture content of 40% in the system. The resulting wet mixture was then transferred to a multi-layer polythene bag equipped with a one-way valve (Rou Duoduo Biotechnology Co., Beijing, China, 20 × 18 cm). After sealing, the polyethylene bag was placed in an incubator with the temperature set at 30 °C to ferment for 3 days. At the end of fermentation, the polysaccharide content and image characteristics of the fermented Glycyrrhiza stems and leaves with different addition amounts of compound probiotics were analyzed, respectively. The optimal amount of compound probiotics was selected for the subsequent fermentation of Glycyrrhiza stems and leaves.

The optimal fermentation time for Glycyrrhiza stems and leaves in laboratory pre-screening is approximately 72 h. To study the entire fermentation process, the fermentation time was extended to 144 h, and a total of 60 samples were collected by removing five bags of fermented licorice stems and leaves from the incubator every 12 h. During sample acquisition, image information was obtained using a laboratory-made computer vision system (CVS). Additionally, 100 g of samples were collected at each fermentation stage for live cell count and pH measurements. Finally, the fermented licorice stems and leaves were dried at 45 °C and crushed for chemical experiments.

### 2.2. Construction of Computer Vision System and Image Collection

In this research, a self-built computer vision system was used to capture images of fermented Glycyrrhiza stem and leaf samples (Figure 1). The image acquisition system consisted of a dark box with non-reflective interior walls, a standardized light source comprising four D65 lamp tubes (Suzhou Xinmei and Instrument Co., Suzhou, China) positioned at a specific distance above the samples for uniform illumination, and a camera lens (HIKROBOT, Hangzhou, China) fixed at the top of the dark box to ensure consistent angle and distance for image capture. The camera settings included an image size of 3024 × 3024 pixels, no zoom or flash mode, ISO sensitivity set at 640, aperture set at f/1.8, exposure time set at 1/100 s, manual operation mode selected, and JPEG image format used with the resolution set at 300 dpi. In this study, ImageJ software (Version 1.52, National Institutes of Health, Bethesda, MD, USA) was utilized to analyze captured images and extract color data from them. This free and open-source software provides various tools for target object selection based on its shape. In this study, the entire region occupied by the Petri dish was selected as the region of interest, and the average RGB values were extracted from it. In addition, color conversion was performed using the previously reported method [16], which involves converting R (red), G (green), and B (blue) values into HSV (hue, saturation, value) and CIE La*b *(lightness, a component, b component) color spaces.

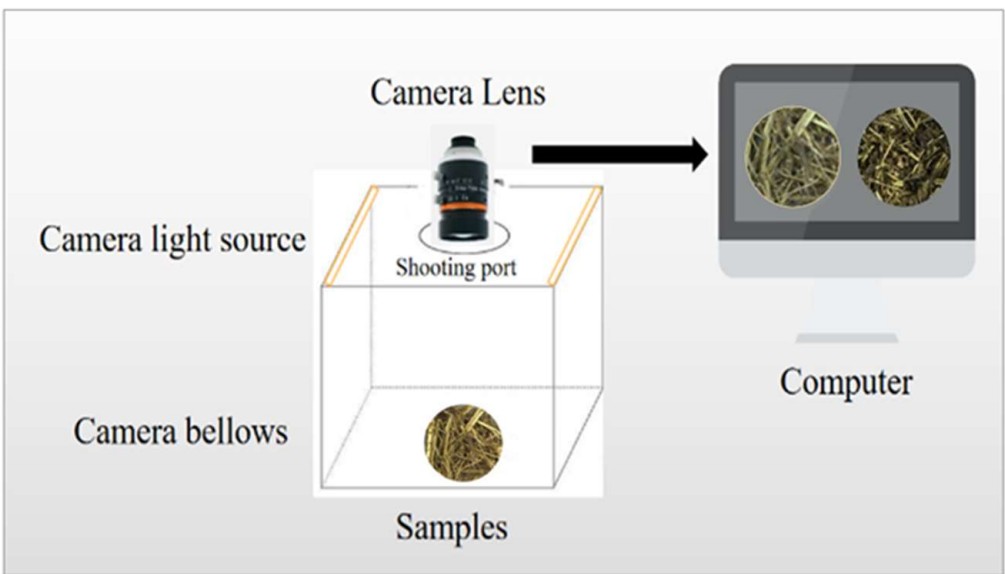

**Figure 1.** Schematic diagram of the computer vision system.

### 2.3. Chemical Analysis

The total polysaccharide content was quantified using the phenol–sulfuric acid method with glucose as the standard [17]. The concentration of TPo was calculated from a standard curve, where Y represents absorbance at 490 nm, and X represents the glucose concentration. The linear regression equation for the standard curve is y = 3.3714x + 0.1876 ($R^2$ = 0.9976). The hot water extraction conditions were performed following previously established methods [18]. A 1.0 g sample of fermented Glycyrrhiza stems and leaves was weighed and subjected to hot water extraction at 80 °C using a ratio of 1:16 (*w/v*) for 30 min. Subsequently, the mixture was centrifuged at 2000× *g* for 30 min, and the resulting supernatant was collected. To prepare the extract solution, 0.5 mL of the supernatant was thoroughly mixed with an equal volume of freshly prepared 5% aqueous phenol solution, followed by the addition of 2.5 mL concentrated sulfuric acid. After thorough mixing, the samples were cooled to room temperature before measuring absorbance at 490 nm.

### 2.4. Microbiological Analysis

Viable cell counts of the fermented Glycyrrhiza stems and leaves were obtained as follows: 5 g of the sample was aseptically transferred from the fermentation bag into sterile test tubes and mixed with 45 milliliters of normal saline solution (0.9% NaCl, *w/v*). A series of 10-fold dilutions were prepared from the homogenate for microbiological analysis, following standard protocols. Each 0.1 mL dilution was inoculated onto corresponding growth media to determine microbial counts. *Lactobacillus plantarum* was cultured on MRS agar plates (MRS broth supplemented with 2% agar) under anaerobic conditions at 37 °C for 48 h, *saccharomycetes* were cultured on wort agar plates (wort broth supplemented with 2% agar) at 28 °C for 36 h, and *Bacillus subtilis* was cultured on nutrient broth agar plates (nutrient broth supplemented with 2% agar) under anaerobic conditions at 37 °C for 48 h.

### 2.5. Statistical Analysis

All test samples were analyzed in triplicate, and the results were expressed as mean ± standard deviation. The data were subjected to analysis of variance (ANOVA) using a Statistical Analysis System (SAS, 2012, SAS Institute, Cary, NC, USA). A ratio of $p < 0.05$ was considered significant, while $p < 0.01$ was considered highly significant.

## 3. Results and Discussion

### 3.1. Polysaccharide Content of Fermented Glycyrrhiza Stems and Leaves with Different Bacterial Additions

Solid-state fermentation of probiotics is an effective way to improve the nutritional quality and bioactivity of food and feed. The right amount of bacterial inoculation and fermentation time is critical to the fermentation process and the quality of the fermented products, as they affect several key parameters such as the reaction rate, oxygen consumption, concentration of fermentation products, and bacterial contamination [19]. In this study, the effects of different inoculum sizes on the polysaccharide content in stems and leaves of Glycyrrhiza were compared during solid-state fermentation to evaluate their characteristics. As shown in Figure 2, when the inoculum size was 0.2%, the fermented Glycyrrhiza stems and leaves had the highest polysaccharide content, which was significantly different ($p < 0.05$). Then, as the inoculum concentration increased, the polysaccharide content decreased. This may be due to the fact that when the inoculum is too high, the nutrients in the culture medium are insufficient to support the growth of fermented strains. As a result, active substances such as polysaccharides are consumed for strain growth, leading to a reduction in the polysaccharide content in fermented Glycyrrhiza stems and leaves [20].

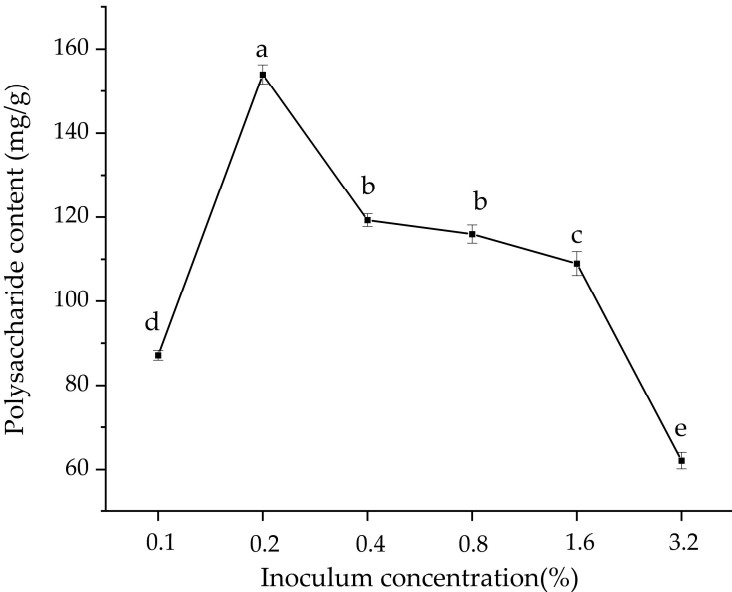

**Figure 2.** Polysaccharide content of fermented Glycyrrhiza stems and leaves with different bacterial additions. Different letters (a, b, c, d, e) above the columns indicated significant differences ($p < 0.05$).

### 3.2. Color Parameters of Fermented Glycyrrhiza Stems and Leaves with Different Bacterial Additions

The color of fermented material is a critical quality parameter that can be used to assess the quality of both raw and fermented materials, ensure feed conforms to quality specifications, and analyze changes in quality resulting from processing, storage, or other factors. In this experiment, to better understand the change in the color of the fresh Glycyrrhiza stem and leaf samples after fermentation with different amounts of bacterial inoculation, we first visually analyzed the original images of the samples, as shown in Figure 3. The results suggest that microorganisms trigger complex glycolysis and polysaccharide reactions during the fermentation process, resulting in color change in certain substances in fermented Glycyrrhiza stems and leaves to a faint yellow tint. As the inoculation amount increases, the color gradually shifts from bright golden yellow to dark brown due to an excess of microorganisms accelerating the fermentation rate, which affects the material's hue. The feature values were extracted from the original images to provide

a more objective evaluation of image information. Color can be defined and measured using a set of three or four coordinates, known as color space. The trends for different color variable spaces, including RGB, HSV, and La*b*, are shown in Figure 4. The values of R, G, and B exhibit a trend of initially increasing and then decreasing, reaching their maximum value at a 0.2% inoculation dose (Figure 4a–c). These results align with the findings on the polysaccharide content, indicating that the inoculation dosage is moderate. In the HSV color space (Figure 4d–f), the H value gradually decreases and reaches its lowest point when the addition amount is 3.2%. The S value increases gradually, while the V value first increases and then decreases. The L* value of a sample indicates its lightness. As the quantity of bacteria added increases, the L value gradually decreases (Figure 4g). The a* and b* values indicate the intensity of red and yellow in the fermented Glycyrrhiza stems and leaves, respectively (Figure 4h,i). As the quantity of bacteria added increases, the red intensity of fermented Glycyrrhiza stems and leaves increases, while the yellow intensity decreases. These results also agree with visual observations.

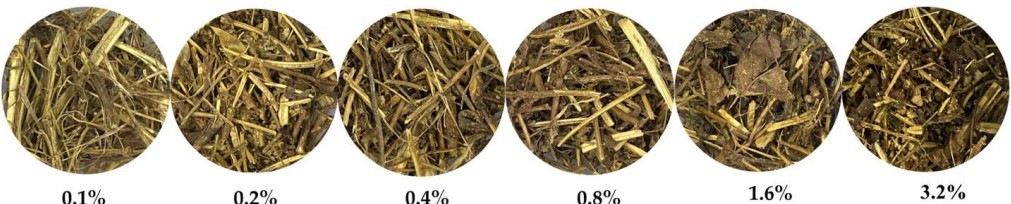

**Figure 3.** Images of fermented Glycyrrhiza stems and leaves with different bacterial additions.

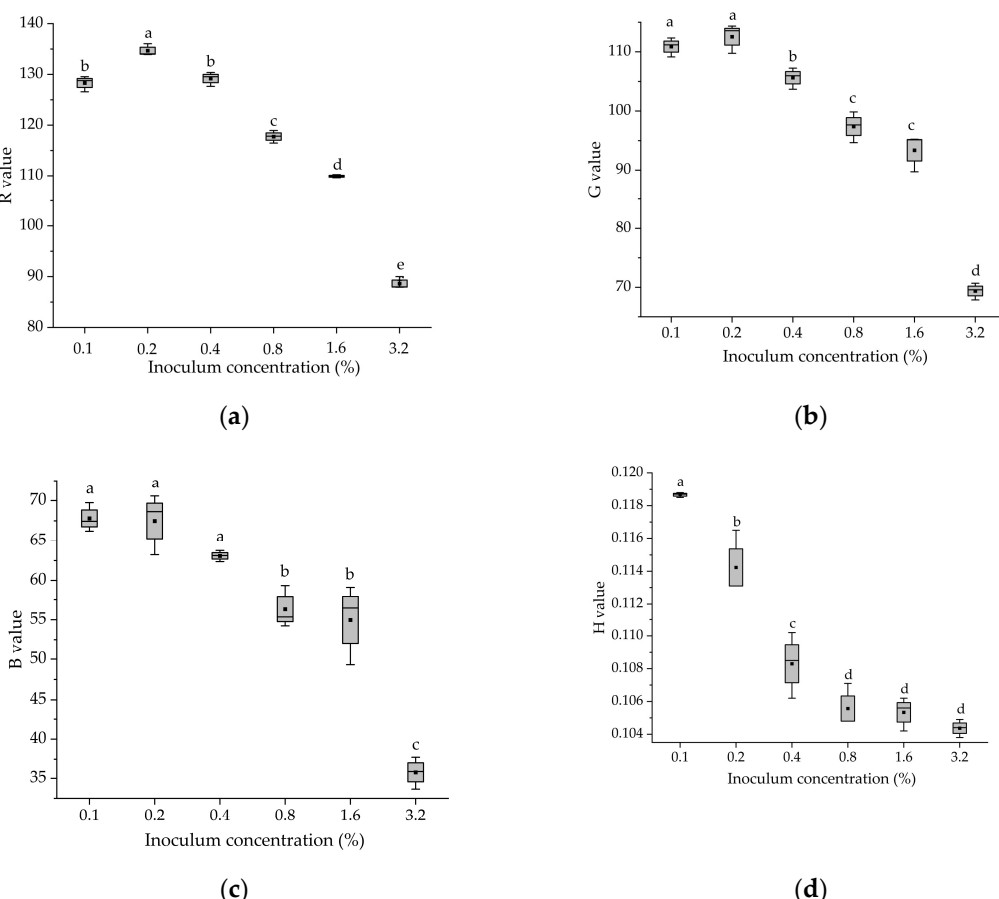

**Figure 4.** *Cont.*

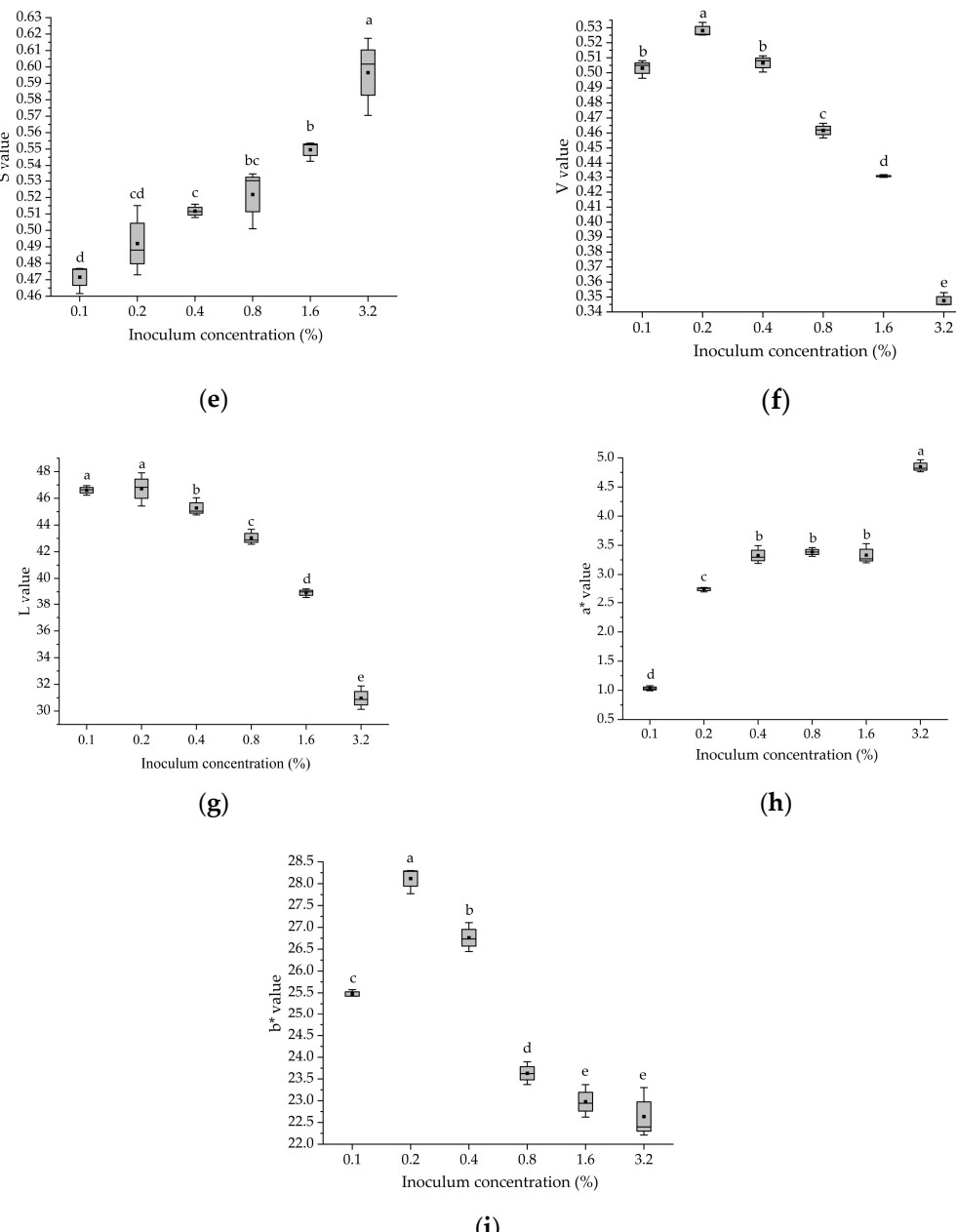

**Figure 4.** Color parameters of fermented Glycyrrhiza stems and leaves with different bacterial additions: (**a**) R; (**b**) G; (**c**) B; (**d**) H; (**e**) S; (**f**) V; (**g**) L; (**h**) a*; and (**i**) b*. Different letters (a, b, c, d, e) above the columns indicated significant differences (*p* < 0.05).

### 3.3. Dynamic Changes in the Viable Counts of the Probiotic Strains and pH Values of Glycyrrhiza Stems and Leaves during Fermentation

Figure 5 depicted the changes in viable counts of probiotic strains throughout fermentation in this study. For fermented foods, the growth and reproduction of colony are related to the fermentation time. With an increase in fermentation time, the colony growth and reproduction will go through four stages: lag phase, logarithmic phase, stationary phase, and stable phase. The colony composition will also change accordingly [21], which will cause changes in the colony vitality, sensory acceptance, and product quality. The initial viable cell counts of probiotic strains were 4.522 log CFU/g (*bacillus subtilis*), 4.497 log CFU/g (*Lactobacillus plantarum*), and 4.402 log CFU/g (*saccharomycetes*). During fermentation, the number of all probiotic strains showed an increasing trend in the fermented Glycyrrhiza stems and leaves. At 24 h, the sample entered the logarithmic phase, *Lactobacillus plantarum*,

*saccharomycetes*, and *bacillus subtilis* rapidly increased, and then, the colony growth entered a stationary phase, maintained a slowly increasing trend, reaching the highest of 9.111 log CFU/g, 6.974 log CFU/g, and 7.096 log CFU/g, respectively. After 96 h, the colony entered a decline phase and began to decline. This may be related to the decrease in nutrients in the sample after long-term fermentation, which is difficult to supply for rapid reproduction of colonies. In this study, the viable accounts of probiotic strains in all fermented samples reached the standard, and an increase in bacterial number indicated that the Glycyrrhiza stems and leaves were utilized by the probiotics for growth. Correspondingly, the pH value of the Glycyrrhiza stems and leaves showed a decreasing trend during the fermentation process (Figure 6). More specifically, the initial pH of 5.22 rapidly decreased to 4.80 within 24 h, which can be attributed to the rapid proliferation of inoculated microorganisms in the Glycyrrhiza stem and leaf matrix, resulting in increased production of lactic acid and other acids. After a period of gentle decline, it rapidly decreased to 4.43 at 72 h and then continued to slowly decrease until it stabilized (at $4.17 \pm 0.02$ at 132 h). This may be due to the accumulation of metabolites slowing down microbial metabolic activities [22].

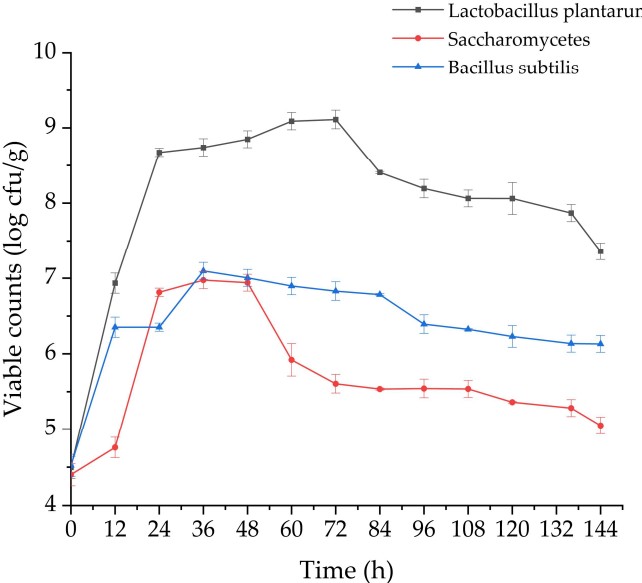

**Figure 5.** Changes in the viable counts of the probiotic strains during the fermentation of Glycyrrhiza stems and leaves.

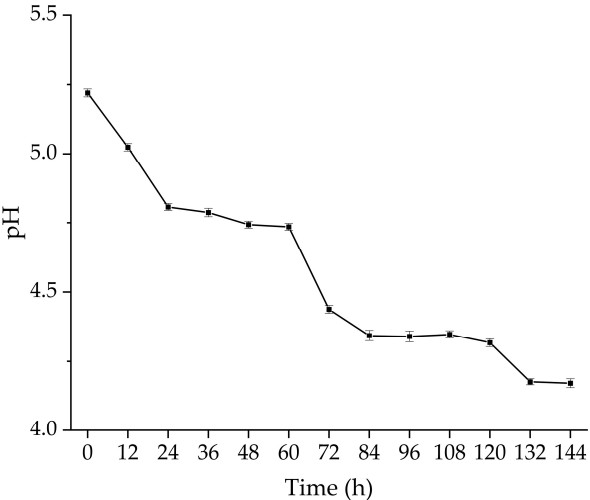

**Figure 6.** The change in pH value during the fermentation of Glycyrrhiza stems and leaves.

### 3.4. Dynamic Changes in Polysaccharide Content of Glycyrrhiza Stems and Leaves during Fermentation

Microbial fermentation utilizes enzyme and fermentation engineering techniques to break down the cellulose and lignin found in the cell walls of Chinese herbal medicines, which then results in the release of their active constituents [23]. Polysaccharides, which are the primary active components found in Glycyrrhiza stems and leaves, serve as crucial indicators for assessing fermentation quality. Figure 7 illustrates the dynamic changes in the polysaccharide content throughout the fermentation process. During the initial fermentation process, there was a significant increase in the polysaccharide content of Glycyrrhiza stems and leaves at 12 h ($p < 0.05$), followed by a decrease at 24 h ($p < 0.05$). The primary cause of polysaccharide consumption was attributed to their utilization by microorganisms for the purpose of cellular proliferation and conversion into organic acids, notably lactic acid [24,25]. At 60 h, there was a significant increase in the polysaccharide content at a high rate ($p < 0.05$), reaching its maximum of 145.56 mg/g at 72 h ($p < 0.05$). This may be attributed to the bacteria's vigorous metabolism and secretion of lignocellulolytic enzymes that degrade lignocellulose, resulting in the production of numerous monosaccharides or polysaccharides. At this time, the fermentation of Glycyrrhiza stems and leaves was moderate. As the fermentation progressed, excessive fermentation was indicated by a significant decrease in polysaccharide content after 108 h ($p < 0.05$). This was also observed by Adetuyi and Ibrahim [26] in the fermentation of Okra Seeds. This may be due to the rapid growth of bacteria, leading to a faster consumption and transformation of polysaccharide, which decreases the total polysaccharide content. Therefore, the fermentation of Glycyrrhiza stem and leaf samples can be classified into preliminary, moderate, and excessive categories based on their polysaccharide content.

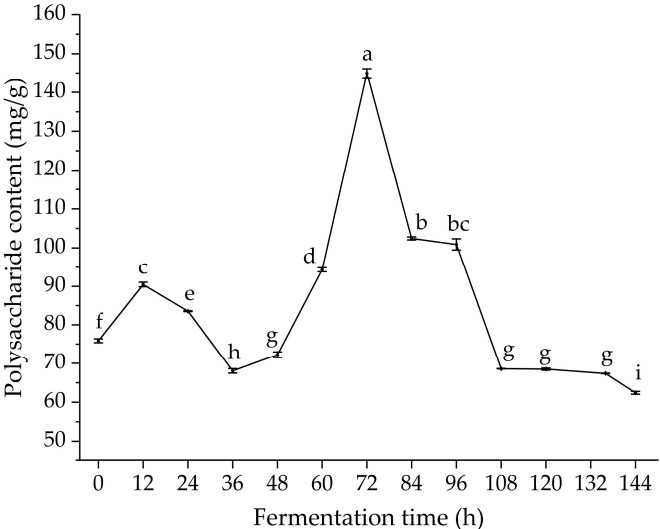

**Figure 7.** Dynamic changes in polysaccharide content of Glycyrrhiza stems and leaves during fermentation. Different letters (a, b, c, d, e, f, g, h, i) above the columns indicated significant differences ($p < 0.05$).

### 3.5. Dynamic Changes in Color Parameters of Glycyrrhiza Stems and Leaves during Fermentation

The original images of fermented Glycyrrhiza stem and leaf samples at different fermentation durations are presented in Figure 8a. At 0 h, the fermented sample of Glycyrrhiza stems and leaves was light yellow overall, but some leaves exhibited a distinct peak green appearance. Since the color change in the Glycyrrhiza stem and leaf samples at each stage of fermentation was not so obvious compared to the change in the whole process, visual enhancement technology was used (Figure 8b). As the fermentation duration increased, the Glycyrrhiza stem and leaf samples changed from light yellow and golden yellow at the beginning to yellowish brown. This may be related to the occurrence of the Mail-

lard reaction, which is a chemical reaction that occurs between reducing sugars' carbonyl groups and amino acid, peptide, or protein amino groups. The color change is facilitated by the production of polymeric products such as melanoidins and advanced glycation end products (AGEs) formed at the advanced stages of MR [12,27].

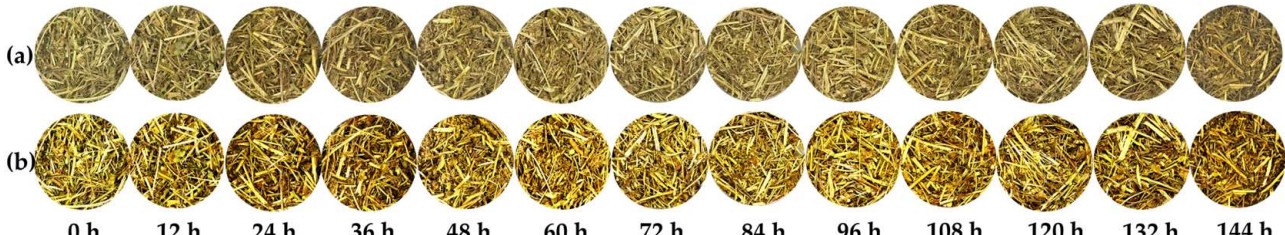

**Figure 8.** Dynamic changes in images of Glycyrrhiza stems and leaves during fermentation: (**a**) original images and (**b**) contrast-enhanced images.

Furthermore, the trends for the different color variable spaces, including RGB, HSV, and Lab, are shown in Figure 9. In the RGB color space, the three variables exhibited dynamic changes as the fermentation duration increased (Figure 9a–c). The values of R, G, and B initially decreased before increasing again, reaching their maximum at 72 h ($p < 0.05$). Conversely, in the HSV color space (Figure 9d–f), H and S variables displayed opposing trends with respect to the fermentation duration, representing the hue and saturation of Glycyrrhiza stem and leaf samples, respectively. In addition, V values showed similar trends to R, G and B values. In the L*a*b* space (Figure 9g–i), the b* values showed monotonic changes during fermentation, while the a* values increased significantly ($p < 0.05$), implying a deepening of the brownness of the Glycyrrhiza stem and leaf samples. To sum up, fermentation led to subtle alterations in all color parameters, and in the final fermentation stage, all color variables decreased sharply to their lowest values except for increases in a* and b* values, which causes the color of fermented licorice stems and leaves to change from light yellow to yellowish brown. Through the analysis of color variables extracted from the images, it became possible to provide a quantitative description of fermentation quality that cannot be achieved through visual evaluation using the naked eye.

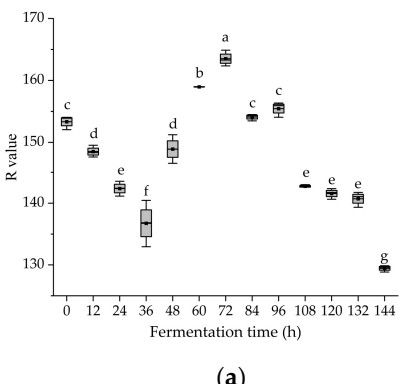

(**a**)

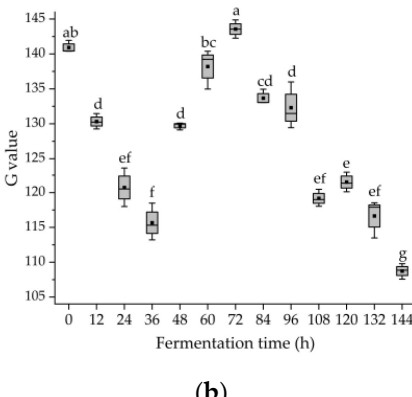

(**b**)

**Figure 9.** *Cont.*

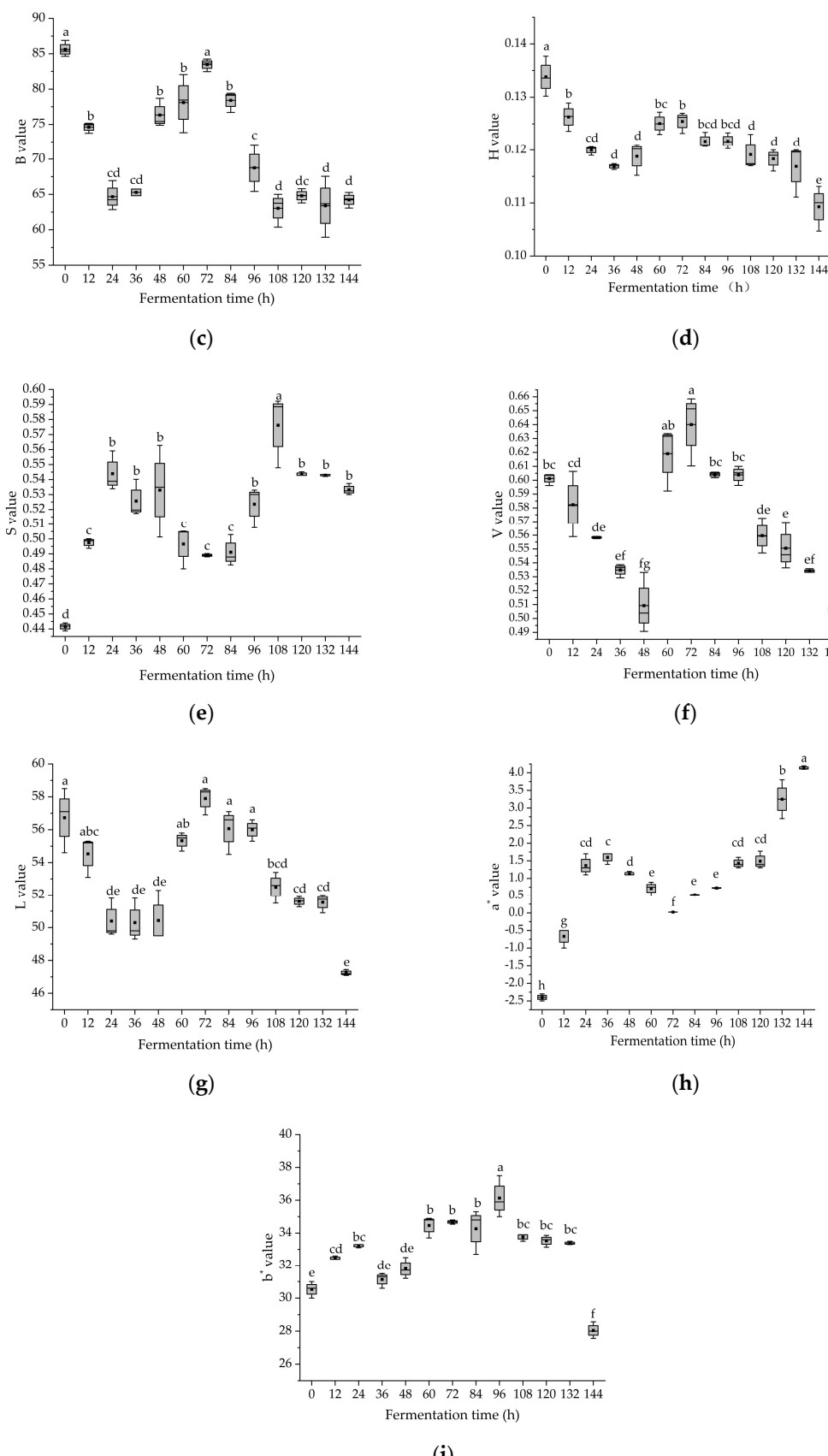

**Figure 9.** Dynamic changes in color variables of Glycyrrhiza stems and leaves during fermentation: (**a**) R; (**b**) G; (**c**) B; (**d**) H; (**e**) S; (**f**) V; (**g**) L; (**h**) a*; and (**i**) b*. Different letters (a, b, c, d, e, f, g, h) above the columns indicated significant differences ($p < 0.05$).

### 3.6. Correlations between Imaging Variables and Quality Indicators

Figure 10 illustrates the correlation between color variables and the polysaccharide content in the fermentation process of Glycyrrhiza stems and leaves. The colors red and blue indicate strong positive and negative correlations, respectively. R, G, S, and V exhibited highly significant positive correlations with the polysaccharide content (coefficients of 0.97, 0.83, 1.00 and 0.98, respectively), while B, a*, and b* showed significant positive correlations with the polysaccharide content (coefficients of 0.65, 0.61, and 0.71, respectively). On the contrary, a significant inverse relationship was observed between L and the polysaccharide content (with a coefficient of −0.62), while H did not display any noteworthy association with the polysaccharide content (r = −0.42). In general, the quality indicators exhibited strong correlations with eight color variables including R, G, B, S, V, L, a*, and b*. These variables may have played a more prominent role in predicting the quality indicators compared to other factors.

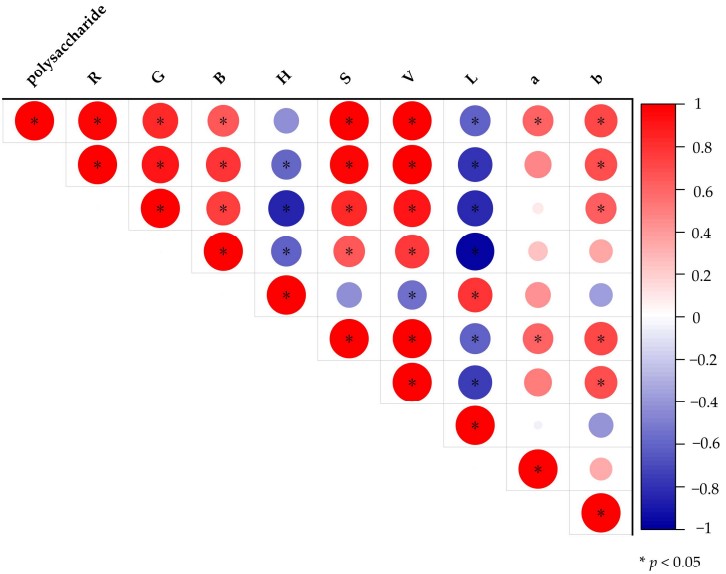

**Figure 10.** Correlations between imaging variables and quality indicators. Heat map of correlation coefficient between quality indicators and color variables.

Principal component analysis (PCA) was conducted utilizing color parameters and the polysaccharide content to emphasize the crucial properties of fermented Glycyrrhiza stems and leaves throughout the fermentation process. Figure 11 displays the scores for the first two principal components, which explain a total variance of 85.72% (PC1 = 70.79% and PC2 = 14.93%). It is evident that all four fermentation stages are almost completely separated. On day 0, the Glycyrrhiza stem and leaf samples were found to be located in a distinct quadrant, far from the other sampling points. This suggests that there were notable alterations in the endogenous metabolites during fermentation [28]. Fermented Glycyrrhiza stems and leaves at hour 12 were distributed in the fourth quadrant, close to the X-axis. Fermented Glycyrrhiza stems and leaves at hour 24–48 were distributed in the second or third quadrants, close to the X-axis. The fermented stems and leaves of Glycyrrhiza at hour 60–96 mainly occupied the first quadrant, while others were present in either the second or third quadrants. From the spatial distribution of the sampling points, the Glycyrrhiza stems and leaves at different fermentation stages exhibited significant differences and were distributed in different regions. This suggests that PCA based on color variables can effectively distinguish the fermentation stages.

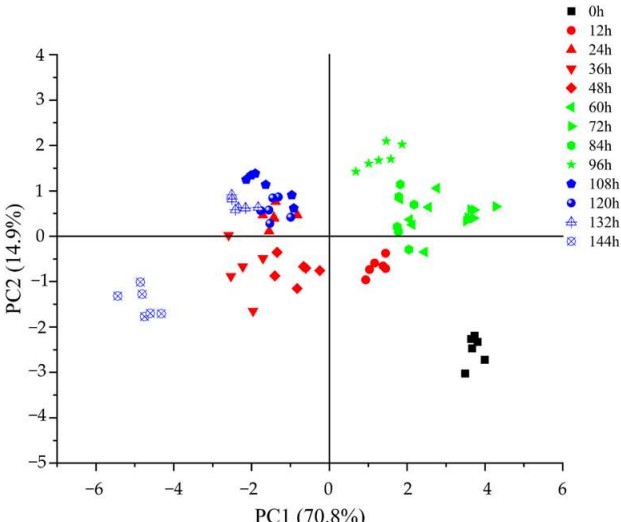

**Figure 11.** Principal component analysis of color parameters and polysaccharide content of fermented Glycyrrhiza stems and leaves. Percentage values in parentheses of PC1 and PC2 indicate the experimental variation explained by each component.

## 4. Conclusions

This paper explored the impact of solid-state fermentation on the polysaccharide content and color parameters of Glycyrrhiza stems and leaves through computer vision chemical analysis. The results indicated that the fermented Glycyrrhiza stems and leaves had the highest polysaccharide content when the inoculum size was 0.2%. Color measuring was successfully used to objectively characterize the fermented Glycyrrhiza stems and leaves with different bacterial additions, resulting in an excellent agreement with visual observations. Throughout the fermentation process, all fermented samples met the standard for viable counts of probiotic strains, and the polysaccharide content reached its peak at 72 h, according to the dynamic change of polysaccharide content, and the fermentation degree can be divided into three categories: preliminary, moderate, and excessive fermentation. The image features and color parameters showed that the Glycyrrhiza stems and leaves samples gradually change from green and light yellow at the beginning to golden and yellow brown. Furthermore, this study revealed a strong correlation between the color and polysaccharide content. The PCA analysis revealed that fermentation stages can be effectively distinguished based on color variables. These findings provide a reference for achieving high-value utilization of Glycyrrhiza stems and leaves, as well as offering new ideas for monitoring and quality control research during the fermentation process.

**Author Contributions:** Conceptualization, J.D. and J.Q.; methodology, J.D.; software, Y.S.; validation, J.D., Y.S. and X.L.; formal analysis, X.A.; investigation, J.D.; resources, X.A.; data curation, J.D., X.A. and J.Q.; writing—original draft preparation, J.D.; writing—review and editing, X.A. and J.Q.; visualization, X.A. and N.L.; supervision, J.Q.; project administration, J.Q.; funding acquisition, X.A. and J.Q. All authors have read and agreed to the published version of the manuscript.

**Funding:** This study was supported by the Major Science and Technology Program of Inner Mongolia Autonomous Region (2020ZD0004 and 2021ZD0023-3) and Research Project of Inner Mongolia Agricultural University (NDKY2021-05).

**Institutional Review Board Statement:** Not applicable.

**Informed Consent Statement:** Not applicable.

**Data Availability Statement:** Not applicable.

**Conflicts of Interest:** The authors declare no conflict of interest.

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
