# Peer review of "Dynamic Changes and Correlation Analysis of Polysaccharide Content and Color Parameters in Glycyrrhiza Stems and Leaves during Fermentation"

_fermentation, doi:10.3390/fermentation9100900_

Round 1
Reviewer 1 Report
Authors described the simple fermentation procedure to visualize the fermented colors of Glycyrrhiza stem and leaves by a camera lens fixed at the top of the dark box. What was the aim of this fermentation? Please described clearly in the abstract and introduction section. Authors emphasied the uses of wastes, Glycyrrhiza stem and leaves, in the fermentation process. What was the final treatment of these fermented solids? Several weaknesses were fond in the MS.
1. There was no continuous stirring during fermentation. What is the method to control the pH and temperature. The bubbles were generally produced druring fermentation. How is the method to improve the interference for the computer vision system.
2. The contamination will happen during the samplings. There was no sterilization procedure before the fermentation procedure.
3. It seemed that maize meal (20%, w/w), soybean meal (10%, w/w), and wheat bran (10%, w/w) were added in the fermentation procedure other than stem and leaves of licorice. It should be answered that the increased polysaccharides (or just simple sugars, monosaccharide or disaccharide) derived from Glycyrrhiza stem and leaves or the basal medium (maize meal, soybean meal and eheat bran). It was advised to perform the same fermentation procedure with the basal medium only to check the changes of polysaccharides and fermentation colors.
4. The materials of stems or leaves of licorice should be described for the cultivation peroids (such as for 2-year or 3-year stems of licorice, or one-year leaves of licorice).
Reviewer 2 Report
The aim of this manuscript is valuable that one model was established to evaluate the quality of the food materials. In the manuscript, some research results also have been given. There are two sides can be improved.
1. Most of all, the research technologies can be improved, for example, to realize computer vision-based analytical chemistry, not just using one camera with some pictures of Glycyrrhiza stems and leaves in Fig. 3 and 8. In fact, as for the nondestructive testing, the data models should be set up further.
2. The figures in the manuscript should be inserted renewedly, such as, Figure 5 with line at right, Figure 5 with lines and black background.
English can be improved, for example, Lines 40-41, what is “has had”?
Author Response
Reviewer #2 Comments:
Point 1: Most of all, the research technologies can be improved, for example, to realize computer vision-based analytical chemistry, not just using one camera with some pictures of Glycyrrhiza stems and leaves in Fig. 3 and 8. In fact, as for the nondestructive testing, the data models should be set up further.
Response: Thanks for your comment. this study aims to investigate 1) the effect of different bacterial additions on the content of polysaccharide and color attributes; 2) dynamic changes in polysaccharide content and color; 3) the correlation relationship between the polysaccharide content and color in Glycyrrhiza stems and leaves during fermentation. This study lays a foundation for future computer vision-based quality identification of fermented Glycyrrhiza stems and leaves. In future research, we will establish a computer vision-based quality identification model based on this foundation.
Point 2: The figures in the manuscript should be inserted renewedly, such as, Figure 5 with line at right, Figure 5 with lines and black background.
Response: Thanks for your comment. The figures have been reinserted into the manuscript.
Comments on the Quality of English Language
English can be improved, for example, Lines 40-41, what is “has had”?
Response: Thanks for your comment. We are sorry for the writing mistakes. We have used MDPI's language editing service to improve the English language quality of the manuscript.
Round 2
Reviewer 1 Report
The responses to the point 3 and the data for Figure 7. These basal media [maize meal (20%, w/w), soybean meal (10%, w/w), wheat bran (10%, w/w) and cellulase (0.5%, w/w)] provide not only for nitrogen sources but also for carbon sources. However, these basal media really contain polysaccharides. It is suggested to perform the polysaccharide changes (Figure 7) by using basal medium.
Reviewer 2 Report
The manuscript has been revised and may be considered for publication.
Author Response
Thanks for your comment.
Round 3
Reviewer 1 Report
Accept in present form.
Author Response
Thanks for your comment.